# Genetic and Metabolic Factors of Familial Dysbetalipoproteinemia Phenotype: Insights from a Cross-Sectional Study

**DOI:** 10.3390/ijms26157376

**Published:** 2025-07-30

**Authors:** Anastasia V. Blokhina, Alexandra I. Ershova, Anna V. Kiseleva, Evgeniia A. Sotnikova, Marija Zaicenoka, Anastasia A. Zharikova, Yuri V. Vyatkin, Vasily E. Ramensky, Elizaveta A. Novokhatskaya, Anna L. Borisova, Svetlana A. Shalnova, Alexey N. Meshkov, Oxana M. Drapkina

**Affiliations:** 1National Medical Research Center for Therapy and Preventive Medicine, Ministry of Healthcare of the Russian Federation, Petroverigsky per. 10, Bld. 3, 101000 Moscow, Russia; alersh@mail.ru (A.I.E.); sanyutabe@gmail.com (A.V.K.); sotnikova.evgeniya@gmail.com (E.A.S.); marija.zaicenoka@gmail.com (M.Z.); azharikova89@gmail.com (A.A.Z.); vyatkin@gmail.com (Y.V.V.); ramensky@gmail.com (V.E.R.); lisabet244@gmail.com (E.A.N.); a.borisova0108@gmail.com (A.L.B.); sshalnova@gnicpm.ru (S.A.S.); meshkov@lipidclinic.ru (A.N.M.); drapkina@bk.ru (O.M.D.); 2Moscow Center for Advanced Studies, 20 Kulakova Str., 123592 Moscow, Russia; 3Faculty of Bioengineering and Bioinformatics, Lomonosov Moscow State University, 1-73, Leninskie Gory, 119991 Moscow, Russia; 4Department of Natural Sciences, Novosibirsk State University, 1, Pirogova Str., 630090 Novosibirsk, Russia; 5Institute for Artificial Intelligence, Lomonosov Moscow State University, 1-73, Leninskie Gory, 119991 Moscow, Russia; 6National Medical Research Center for Cardiology, 3-ya Cherepkovskaya Str., 15A, 121552 Moscow, Russia; 7Research Centre for Medical Genetics, 1 Moskvorechye Str., 115522 Moscow, Russia; 8Department of General and Medical Genetics, Pirogov Russian National Research Medical University, 1 Ostrovityanova Str., 117997 Moscow, Russia

**Keywords:** familial dysbetalipoproteinemia, hyperlipoproteinemia type III, *APOE*, ε2/ε2, remnant lipoprotein, polygenic hypertriglyceridemia, polygenic risk score, additive risk, metabolic factors, cardiometabolic index

## Abstract

Familial dysbetalipoproteinemia (FD) is a prevalent and highly atherogenic hyperlipoproteinemia associated with the ε2/ε2 *APOE* genotype or rare *APOE* variants. The contributions of additional genetic and clinical factors to the FD phenotype remain unclear. We investigated these factors in both autosomal recessive and autosomal dominant forms of FD. Targeted (*n* = 4666) and exome (*n* = 194) sequencing were used to identify the ε2/ε2 *APOE* genotype or rare FD-causative *APOE* variants. Twenty-four lipid-related genes and forty variants included in a polygenic risk score for hypertriglyceridemia (HTG) were analyzed. FD was defined by the presence of FD variants and triglycerides (TG) ≥ 1.5 mmol/L (main study group). The comparison group consisted of patients with FD variants but TG < 1.5 mmol/L. Univariable and multivariable regression analyses were performed. A total of 71 unrelated subjects were identified (45.1% male, median age 50 years). FD was diagnosed in 52 patients, while 19 had FD variants only. Age (*p* = 0.019), elevated polygenic risk for HTG (*p* = 0.001), and the presence of metabolic syndrome components (*p* = 0.014) were independently associated with the FD phenotype. TG levels were significantly associated with polygenic burden (0.05 mmol/L per percentile), the presence of additional rare lipid-related variants (7.0 mmol/L), and glucose metabolism disorders (3.62 mmol/L), together explaining 30% of TG variance in cross-validated model. These results highlight the interplay of genetic and metabolic factors in FD development and support the integration of HTG genetic risk scores and metabolic control into personalized FD management.

## 1. Introduction

Familial dysbetalipoproteinemia (FD), also known as type III hyperlipoproteinemia (OMIM #617347), is a genetically determined lipid disorder primarily linked with *APOE* gene variants [1,2,3,4]. In recent years, interest in FD has grown because of several key insights.

First, the widespread implementation of next-generation sequencing (NGS) has greatly enhanced our understanding of the genetic causes of this hyperlipidemia. It is now recognized that over 90% of FD cases are associated with the ε2/ε2 *APOE* genotype and inherited in an autosomal recessive (AR) manner [5,6], whereas the remaining cases are linked to rare pathogenic *APOE* variants with autosomal dominant (AD) inheritance [1,2,7,8]. Previously, the pathogenicity of rare *APOE* variants in FD was unclear, and comprehensive genetic testing was often unavailable. As a result, most genotype-phenotype association studies focused specifically on ε2/ε2 cases, limiting findings to the AR form of FD. In a previous study of a Russian patient cohort, we described the pathogenicity of several rare *APOE* variants and genotypes associated with FD [1]. These findings open new opportunities to expand research on both AR and AD forms of FD, specifically enabling detailed phenotypic characterization of this disease.

Second, FD is one of the most atherogenic hyperlipidemias, alongside familial hypercholesterolemia (FH). It is characterized by the accumulation of cholesterol-enriched remnant lipoproteins, which promotes early-onset coronary [9,10,11,12] and peripheral atherosclerosis [9,13]. Third, our understanding of FD prevalence has evolved significantly. Once considered a rare disease, FD is now recognized as a prevalent hyperlipidemia, with prevalence estimates ranging from 0.2% to 0.7% [14,15,16,17], comparable to that of FH [14].

Notably, both AD and AR forms of FD share a unique feature that distinguishes them from FH: the presence of FD-associated genetic variants alone is insufficient to cause the disease. Additional factors, either genetic modifiers beyond *APOE* variants or clinical factors, are necessary for FD development. However, the specific factors that contribute to the development of FD remain incompletely understood. Among the better-studied clinical factors are metabolic disorders such as overweightness and obesity [6,18], insulin resistance [6,19,20], and diabetes mellitus [6,11]. The role of additional genetic factors is less well studied but may include pathogenic variants in lipid-related genes beyond *APOE*, such as *APOA5*, *APOC3*, and *LPL* [21,22], as well as an elevated polygenic risk burden [23,24].

The interaction between FD-associated genetic variants and additional modifying factors contributes to the clinically heterogeneous FD phenotype, complicating timely diagnosis [3,9,11,12,18]. A better understanding of the genetic and clinical factors to the FD phenotype may be essential for developing personalized management strategies, including preventing FD onset and reducing atherosclerotic cardiovascular (CV) risk in carriers of FD-associated variants [12,18].

Therefore, the aim of this study was to investigate the frequency and contribution of both genetic and clinical factors to the FD phenotype.

## 2. Results

A graphical summary of the study process is presented in Figure 1.

### 2.1. Genetic Factors

Genetic data from 4860 subjects were analyzed to identify the ε2/ε2 *APOE* genotype and rare *APOE* variants associated with FD.

A total of 71 unrelated subjects were identified. Among them, 49 carried the ε2/ε2 genotype, which is linked to the AR form of FD. Rare causative *APOE* variants associated with the AD form were identified in 22 patients. Of these, 19 were heterozygous carriers of rare *APOE* variants, including p.Glu63ArgfsTer15 (*n* = 1), p.Gly145AlafsTer97 (*n* = 1), p.Lys164SerfsTer87 (*n* = 1), p.Arg154Cys (*n* = 10), p.Glu230Lys (*n* = 1), and p.Gly145Asp-ε2/ε3 (*n* = 5). Additionally, three patients carried the rare ε2/ε1 genotype, combining the heterozygous p.Gly145Asp variant with the homozygosity for p.Arg176Cys.

Applying a triglyceride (TG) threshold of ≥1.5 mmol/L [14], FD was diagnosed in 52 patients (the main study group), representing 73.2% of all carriers of FD-associated variants. Among them, 37 (71.2%) had the AR form and 15 had the AD form (*p* = 0.003). Within the AD subgroup, the most frequent variant was p.Arg154Cys, identified in 53.3% of cases. The ε2/ε1 genotype was present in 20.0% of patients. The remaining 19 subjects, carriers of either the ε2/ε2 genotype or rare FD-causative *APOE* variants, with TG levels <1.5 mmol/L, were assigned to the comparison group. The distribution of ε2/ε2 versus rare *APOE* variants did not differ significantly within this group (*p* = 0.359) (Figure 2). The frequency of rare *APOE* variants also did not differ significantly between the FD and comparison groups (odds ratio (OR) = 0.7; 95% confidence interval (CI): 0.2–2.5; *p* = 0.569).

#### 2.1.1. Pathogenic Variants in Lipid-Related Genes Beyond APOE

Among FD patients, 9.6% carried additional pathogenic variants in lipid-related genes beyond *APOE*, compared with 5.3% in the comparison group (*p* = 1.0). Specifically, six additional pathogenic variants across three different lipid-related genes were identified within the FD group. Two patients carried both FD-associated and FH-associated variants. Furthermore, three patients had variants linked to familial chylomicronemia syndrome: one was compound heterozygous for two likely pathogenic variants in *LPL*, another was homozygous for a likely pathogenic variant in *LMF1*, and the third was heterozygous for a likely pathogenic variant in *LPL* (Table 1).

#### 2.1.2. Polygenic Risk

Polygenic hypertriglyceridemia (HTG) was significantly more frequent among FD patients than among subjects with FD variants: 40.0% vs. 5.6%, respectively (OR = 14.9; 95% CI: 1.8–706.5; *p* = 0.003). Although the CI was wide, reflecting the small number of high-polygenic-risk cases in the FD-variant group, the association remained statistically significant. In contrast, most patients with FD variants (61.1%) had a low polygenic contribution to TG levels. The distribution of HTG polygenic risk score (PRS) percentiles also differed markedly between groups: 69 (45; 89) in FD patients vs. 39 (26; 56) in the FD variants group. The median difference (ΔMe) was 25 percentiles (95% CI: 7–41); *p* = 0.009, indicating a significantly higher polygenic burden of HTG in FD patients (Figure 3).

Conversely, polygenic hypercholesterolemia was infrequent in both groups: 6.0% in FD patients vs. 10.5% in the FD variants group (*p* = 0.850). The medians of hypercholesterolemia PRS percentiles were also comparable: 2 (1; 8) in FD patients vs. 3 (1; 24) in the comparison group (*p* = 0.598).

### 2.2. Clinical Factors

Physical signs were assessed in 32 patients with FD. Among them, 28.1% had xanthomas. Specifically, cutaneous xanthomas were observed in 25.0%, and tendon xanthomas in 12.5% of patients. Three patients had both cutaneous and tendon xanthomas (Figure 4).

Data on pancreatitis were available for 32 FD patients. Among them, four (12.5%) reported a history of acute pancreatitis. Notably, two of them were also carriers of likely pathogenic variants in the *LPL* gene.

Table 2 shows the clinical and biochemical data of FD patients compared with those with FD variants.

Patients with FD were older and exhibited several metabolic features compared with subjects with FD variants. Nearly all FD patients (90.4%) were overweight or obese, with a significantly higher body mass index (BMI) (Table 2). The frequency of obesity was higher among FD patients than among those with FD variants: 50.0% vs. 21.1%, respectively (OR = 3.7; 95% CI: 1.0–17.3; *p* = 0.033). Among FD patients with obesity, 84.6% had class I, 11.5% had class II, and 3.8% had class III obesity. Abdominal obesity was also more common in FD patients than in those with FD variants: 78.6% vs. 31.1%, respectively (OR = 7.7; 95% CI: 1.9–36.5; *p* = 0.001). FD patients also had a higher waist-to-height ratio (WHtR) (*p* = 0.001), with clinically significant values observed in both men (0.6 (0.5; 0.6)) and women (0.6 (0.6; 0.6)). A greater proportion of FD patients had a combination of metabolic syndrome components, and their cardiometabolic index (CMI) was significantly higher than in subjects with FD variants. This difference was observed in both men—median 2.420 (1.420; 5.210) vs. 0.565 (0.453; 0.657), ΔMe 1.854 (95% CI: 0.837–4.713); *p* < 0.001—and women—median 2.100 (1.000; 4.340) vs. 0.205 (0.155; 0.289), ΔMe 1.841 (95% CI: 0.796–4.109); *p* < 0.001.

In addition, FD patients had an early onset of coronary heart disease (the median age was 42 years) and had a significantly higher frequency thereof (Table 2).

Low-density lipoprotein cholesterol (LDL-C) levels were higher in FD patients (median 2.95 mmol/L), while high-density lipoprotein cholesterol (HDL-C) levels were lower (median 1.03 mmol/L) (Table 2). Among men with FD, the median HDL-C level was 1.02 mmol/L (0.81; 1.15), which is clinically borderline, whereas in women, HDL-C levels were significantly below the clinical cutoff, with a median of 1.09 mmol/L (0.87; 1.33).

Patients with FD exhibited moderately elevated TG levels (median 4.47 mmol/L). Most patients (73.1%) had TG levels between 1.7 and <10.0 mmol/L, while severe HTG (TG ≥ 10.0 mmol/L) was observed in 21.2% of FD cases. TG levels in FD patients ranged from 1.54 mmol/L to 26.28 mmol/L. Among patients with AD FD, TG levels were higher than among those with AR FD: median 9.51 mmol/L vs. 4.03 mmol/L, respectively; ΔMe 2.60 (95% CI: 0.02–7.18); *p* = 0.046 (Figure 5).

Furthermore, FD patients had higher remnant cholesterol levels (median 1.87 mmol/L), which is a feature of type III hyperlipoproteinemia (Table 2).

Among FD patients, 98.1% (95% CI: 89.7–100) had at least one additional factor for FD development. A single such factor was identified in 42.3% of patients (95% CI: 28.7–56.8), with increased body weight being the most frequent (present in 81.8% of these cases). More than half of FD patients (55.8%; 95% CI: 41.3–69.5) had at least two additional factors. Specifically, two factors were identified in 46.2% of patients (95% CI: 32.2–60.5), while 9.6% had three factors (95% CI: 3.2–21.0). Notably, increased body weight was present in all combinations of factors (95% CI: 88.1–100) and was most frequently combined with polygenic HTG (62.1% of all combinations; 95% CI: 42.3–79.3) (Figure 6).

### 2.3. Independent FD Phenotype Factors

Univariable Firth logistic regression revealed significant associations between FD and polygenic HTG, age, BMI, obesity, waist circumference (WC), a 0.1-unit increase in WHtR, and the presence of metabolic syndrome components (Figure 7A). In both multivariable models, polygenic HTG remained an independent factor associated with FD. In multivariable model I (Figure 7B), which was adjusted for sex, age, BMI, and glucose metabolism disorders, and in multivariable model II (Figure 7C), which was adjusted for sex, age, and the presence of metabolic syndrome components, a 1-percentile increase in TG PRS was associated with a 3.8% higher likelihood of FD. The presence of metabolic syndrome components was also associated with FD (Figure 7C). Although glucose metabolism disorders were statistically significant predictors of FD in multivariable model I (*p* < 0.05), the wide CI, which included 1.00, limits conclusions regarding their contribution to the FD phenotype (Figure 7B).

We also conducted linear regression analysis to assess the associations between FD phenotype-related factors and TG levels. In univariable analysis, the presence of additional rare variants in lipid-related genes was significantly associated with higher TG levels (β = 6.40; standard error = 1.70; 95% CI: 3.01–9.79; *p* < 0.001; R^2^ = 0.17). Glucose metabolism disorders were also associated with elevated TG levels (β = 3.41; standard error = 1.67; 95% CI: 0.09–6.73; *p* = 0.04; R^2^ = 0.06). The full results of the univariable analysis for all predictors are presented in Appendix A.

The final multivariable model included predictors that best explained the variance in TG levels (Table 3). This model was statistically significant and, after fivefold cross-validation, explained 30% of the TG variance.

### 2.4. Treatment Targets in FD Patients

Half of the FD patients were receiving lipid-lowering therapy (LLT). Most of them (61.5%) were treated with statins, while 7.7% received fenofibrate. Only eight patients (30.8%) received combined LLT. Of these, 50.0% were prescribed statin plus ezetimibe, and 37.5% received statin plus fenofibrate. One patient (12.5%) was treated with triple LLT consisting of a statin, ezetimibe, and fenofibrate. Among all patients receiving statins, high-intensity therapy was the most frequent (87.5%). Notably, no FD patients were prescribed PCSK9 inhibitors.

In patients receiving LLT, non-HDL-C levels were 3.29 mmol/L (2.41; 5.63), and TG levels were 2.52 mmol/L (1.80; 4.90). Only 30.4% of these patients achieved the target non-HDL-C levels < 2.6 mmol/L, and just 20.8% achieved the target TG levels < 1.7 mmol/L.

## 3. Discussion

This study provides a cross-sectional analysis of genetic and clinical factors, primarily metabolic, contributing to the development of the FD phenotype, with a focus on their independent associations with FD using multivariable statistical models. A key strength of the study is the inclusion of both AR and AD forms of FD, combined with comprehensive genetic analysis. Additionally, this is the first study to present such data in a Russian patient cohort.

### 3.1. Polygenic HTG

By analyzing the PRS composed of 40 variants associated with TG levels, we observed a high polygenic burden of HTG among FD patients. Specifically, 40.0% of FD patients had polygenic HTG, with a median PRS percentile of 69—significantly higher than in subjects with FD variants. Importantly, in multivariable regression models, polygenic HTG was identified as an independent factor of the FD phenotype. Furthermore, each increase in PRS percentile was associated with a 0.05 mmol/L rise in TG levels, even after adjusting for additional variants in other lipid-related genes and for glucose metabolism disorders.

Several recent studies have also highlighted the polygenic contribution to TG levels as a factor of FD phenotype. In the study by Pieri 2023, a weighted PRS based on 107 TG-raising variants contributed additively to the FD phenotype, which was defined as the presence of the ε2/ε2 *APOE* genotype and TG levels > 3.0 mmol/L [24]. Similarly, in the cohort study by Šatný 2023 (101 FD patients vs. 80 patients with the ε2/ε2 *APOE* genotype), unweighted TG PRSs constructed from two and fifteen variants, respectively, were significantly associated with FD development. The odds of developing FD were more than threefold higher with the two-variant score and nearly sevenfold higher with the fifteen-variant score [23].

Despite differences in FD phenotype definitions and the number of variants included in the TG PRS, the results of our study also emphasize the role of polygenic HTG as a key “second hit” promoting the FD phenotype. Integrating TG-PRS into personalized FD management, particularly in specialized lipid clinics, may facilitate earlier identification of patients at higher risk for FD [23,24]. However, it is important to interpret PRS in combination with other genetic and metabolic factors that may have additive effects on the FD phenotype [24]. Furthermore, not all FD patients develop atherosclerotic CV diseases. Both traditional CV risk factors and emerging genetic markers, such as the PRS for coronary heart disease, may contribute to the heterogeneous FD phenotype [12]. A comprehensive precision approach may provide the greatest benefits for optimizing personalized management of FD.

Moreover, based on our findings that FD patients have a high polygenic burden for HTG, prospective studies focusing on PRS among FD variant carriers could provide valuable insights into the CV disease burden within this patient cohort.

### 3.2. Rare Additional Genetic Variants Beyond APOE

In our study, the frequency of additional rare pathogenic variants in lipid-related genes beyond *APOE* was similar between patients with and without FD. Moreover, the additional presence of these variants was not independently associated with the FD phenotype in either univariable or multivariable logistic regression models. However, their presence significantly modulated TG levels. After adjustment for other factors, the presence of these variants was associated with a 7.0 mmol/L increase in TG levels, indicating a clinically meaningful effect size. Overall, 9.6% of FD patients carried an additional pathogenic variant associated with genetic hyperlipidemia, including variants linked to familial chylomicronemia syndrome, which is characterized by severe HTG. In particular, the Marmontel 2023 study demonstrated the additive effect of deleterious *APOE* variants on LDL-C levels and the risk of premature atherosclerotic CV disease in patients carrying both *LDLR* and *APOE* variants compared with *LDLR* carriers alone (*p* = 0.027 and *p* = 0.026, respectively) [25].

### 3.3. Metabolic Factors

In our study, patients with FD were older and had a higher frequency of both general and abdominal obesity compared with subjects with FD variants. They also demonstrated significantly elevated BMI, WC, and WHtR, along with increased CMI. Additionally, FD patients more frequently had a combination of metabolic syndrome components.

Although only a few studies have directly compared FD patients to those with FD variants, all, including ours, have highlighted the key role of metabolic burden in the development of the FD phenotype [12,18]. In the Paquette 2024 study, FD patients had significantly higher BMI (median 28.5 (26.0; 31.6) kg/m^2^), WC (mean 96.3 ± 12.7 cm), and frequency of diabetes (11.0%) compared with ε2/ε2 carriers without FD (*p* < 0.0001) [15]. These findings for BMI and WC are comparable to our results. While glucose metabolism disorders alone did not differ significantly between groups in our study, FD patients more frequently had multiple metabolic syndrome components, which included diabetes. In the prospective part of the Heidemann 2021 study, higher BMI (1.2-fold increase in odds per 1 kg/m^2^), WC (1.3-fold per 5 cm), and the presence of non-TG metabolic syndrome (4.4-fold increased odds) were independently associated with FD development [18]. Similarly, in our multivariable logistic regression analysis, age (1.1-fold increase in odds per year) and metabolic syndrome components (6.1-fold increased odds) were independently associated with the FD phenotype after adjustments. Other metabolic factors, such as BMI or glucose metabolism disorders, lost significance after adjustments. Furthermore, the presence of glucose metabolism disorders was independently associated with TG levels.

As an additional indicator of metabolic burden, we included CMI in the group comparison analysis. CMI is a robust marker of hyperglycemia and diabetes risk [26], and it was assessed in FD patients for the first time. It was excluded from the regression analysis, as it is a composite index that incorporates TG levels. Notably, CMI has not yet been studied in Russia, and population-specific cutoffs for clinical interpretation have not been established. Nonetheless, CMI may serve as a promising tool for identifying metabolic risk in FD variant carriers and requires further investigation.

It is important to note that the high prevalence of metabolic factors contributes to the accumulation of cholesterol-enriched remnant lipoproteins and the development of the atherogenic FD phenotype. In our study, the median age of coronary heart disease onset was 42 years, highlighting the early onset of CV disease in FD patients, as we previously reported [9]. Moreover, the majority of FD patients in the present study did not reach recommended targets for non-HDL-C and TG levels, consistently with results from the Koopal 2015 study [11]. Poor management of FD-related risk factors, delayed diagnosis, and suboptimal LLT contribute to a persistently elevated CV risk burden in these patients [11].

## 4. Materials and Methods

### 4.1. Sampling

This study is a continuation of our previous research on FD [1,9,14] and involved analysis of samples from the Biobank of the National Medical Research Center (NMRC) for Therapy and Preventive Medicine (Moscow, Russia) [27] genotyped by January 2025. For this study, genetic data from targeted (*n* = 4666) and exome (*n* = 194) sequencing were used. These genetic datasets included the following data, as previously described:Population samples from the “Epidemiology of Cardiovascular Diseases and Risk Factors in Regions of the Russian Federation” (ESSE-RF) study, a cross-sectional study conducted from 2012 to 2013 across 13 regions of Russia [28]. The current study included representative samples from the Ivanovo region (*n* = 1858) [1] and the Vologda region (*n* = 880) [29], as well as single subjects from other regions (*n* = 167), and participants from the ESSE-FH-RF study with a clinical diagnosis of definite or probable heterozygous FH (*n* = 119) [30];Subjects (the Russian patient sample (RPS)) with diverse chronic noncommunicable diseases, including lipid metabolism disorders, whose blood samples were collected at the NMRC for Therapy and Preventive Medicine Biobank (*n* = 1836) [27].

For the current analysis, we included unrelated probands aged 18 years or older with either the ε2/ε2 *APOE* genotype or rare *APOE* variants previously reported as pathogenic for FD [1]. Exclusion criteria included missing TG level values.

The FD group included both AR and AD forms. AR FD was defined by the presence of the ε2/ε2 *APOE* genotype and TG levels ≥ 1.5 mmol/L, as previously described [14]. AD FD was defined by the presence of rare *APOE* variants causative of FD [1] and TG levels ≥ 1.5 mmol/L. Patients with the ε2/ε1 *APOE* genotype were also included. The comparison group comprised subjects with FD genetic variants but with TG levels < 1.5 mmol/L.

### 4.2. Clinical and Biochemical Data

The current work used retrospective clinical data from all the above-mentioned studies, which were previously described [1]. In this study, we included additional metabolic parameters beyond BMI. WC was measured at the midpoint between the inferior edge of the costal border and the iliac crest on the midaxillary line using an inelastic tape measure (*n* = 58). Abdominal obesity was considered as a WC > 94 cm for men and >80 cm for women [31]. WHtR was calculated by dividing WC (cm) by height (cm), considering a value > 0.5 to be clinically significant [32]. CMI was calculated as [TG (mmol/L)/HDL-C (mmol/L)] × WHtR [26]. Glucose metabolism disorders were defined as clinical diagnoses of either diabetes mellitus, impaired glucose tolerance, or impaired fasting glucose. Metabolic syndrome components included at least two of the following: hypertension, BMI ≥ 30.0 kg/m^2^, or glucose metabolism disorders.

Retrospective lipid levels, including total cholesterol (TC), LDL-C, HDL-C, and TG, were reported in mmol/L. These lipid levels were previously measured using the Abbott Architect C-8000 system (Abbott Laboratories, North Chicago, IL, USA). The current study presented pretreatment lipid levels. In the ESSE-RF study, including ESSE-FH-RF, LDL-C levels were determined directly. LDL-C levels from RPS were presented only for patients with TG levels < 4.5 mmol/L or for whom LDL-C was directly measured. The type and volume of LLT were also analyzed. For patients on regular LLT (*n* = 11), pretreatment LDL-C and TG levels were estimated using a correction factor for LLT from a local study [33], and the statin dose was converted to an equivalent atorvastatin dose. TC and HDL-C levels were not recalculated in this case. Non-HDL-C was calculated as TC minus HDL-C, reported in mmol/L and presented only for patients without regular LLT. Remnant cholesterol was estimated as TC minus HDL-C minus LDL-C and provides an approximate measure of the cholesterol content in remnant lipoproteins, including very-low-density and intermediate-density lipoproteins [11]. Lp(a) levels were assessed using the Tokyo Boeki Sapphire-400 analyzer (Tokyo, Japan) and reported in mg/dL.

We classified patients with FD as having at least a high CV risk. Treatment targets for non-HDL-C (<2.6 mmol/L) and TG (<1.7 mmol/L) were defined using the 2019 ESC/EAS guidelines [31].

### 4.3. Genetic Analysis

#### 4.3.1. NGS

Blood samples were stored at −32 °C in the Biobank of the NMRC for Therapy and Preventive Medicine [27]. NGS was performed using exome and custom target panel designs with two platforms: NextSeq 550 (Illumina, San Diego, CA, USA) and Ion S5 (Thermo Fisher Scientific, Waltham, MA, USA), as previously described [1]. The set of 24 genes associated with dyslipidemia (*ABCA1*, *ABCG5*, *ABCG8*, *ANGPTL3*, *APOA1*, *APOA5*, *APOB*, *APOC2*, *APOC3*, *APOE*, *CETP*, *GPD1*, *GPIHBP1*, *LCAT*, *LDLR*, *LDLRAP1*, *LIPC*, *LIPI*, *LMF1*, *LPL*, *PCSK9*, *SAR1B*, *STAP1*, *USF1*) was analyzed [1]. Sanger sequencing was performed using the Applied Biosystem 3500 Genetic Analyzer (Thermo Fisher Scientific, Waltham, MA, USA) following the manufacturer’s protocol.

#### 4.3.2. Bioinformatic Analysis and Clinical Interpretation

All steps of the bioinformatic analysis were previously described [1]. For the current study, principal component analysis (PCA) was conducted on individual genotypes using the Hail library v.0.2.83-b3151b4c4271. Population cluster identification was inferred by projection onto reference samples from the 1000 Genomes Project Phase 3. Variants with a minor allele frequency < 5% were excluded from the PCA analysis, and linkage disequilibrium pruning was performed with R^2^ = 0.2. All analyzed samples from the final study cohorts belonged to the European cluster on the PCA plot; therefore, no outliers were removed.

Common *APOE* genotypes (ε3/ε3, ε4/ε4, ε2/ε2, ε2/ε3, ε3/ε4, ε2/ε4) were identified using p.Arg176Cys (rs7412) and p.Cys130Ar (rs429358) variants, as described previously [14]. For the present study, we selected patients with rare *APOE* variants previously reported as causative for FD [1], based on the ACMG/AMP 2015 guidelines [34]. Information on the cis- or trans-position of *APOE* variants was not available for evaluation. Therefore, the ε2/ε1 genotype was defined as the presence of the rare heterozygous p.Gly145Asp (rs267606664) *APOE* variant combined with homozygosity for p.Arg176Cys.

Clinical interpretation for *LDLR* variants was based on the Clinical Genome Resource guidelines for *LDLR* variant classification [35]. For other lipid-related genes included in the analysis, interpretation was based on the ACMG/AMP2015 guidelines [34]. Only pathogenic and likely pathogenic variants were reported in this study.

### 4.4. Polygenic Risk Score

PRS was calculated using the β-coefficients from the original study [36], which previously demonstrated significant associations with TG levels (40 variants) and LDL-C levels (57 variants) in the population from the European part of Russia [29] (Appendix A). In the present study, the PRS of the participants was compared with that of 1652 individuals from the population-based ESSE-Ivanovo sample. The *APOE* genotype was included in the final genetic score for LDL-C levels, with weights assigned as follows: −0.9 for ε2/ε2, −0.4 for ε2/ε3, −0.2 for ε2/ε4, 0 for ε3/ε3, 0.1 for ε3/ε4, and 0.2 for ε4/ε4 [37]. Polygenic HTG or hypercholesterolemia were defined as a weighted PRS > 80th percentile (high PRS), while a low PRS was defined as a weighted PRS < 50th percentile. PRS for TG and LDL-C were calculated for all study participants except for three and two patients, respectively. These patients were excluded from PRS calculations because of limitations of the exome sequencing design, which did not capture PRS variants located in noncoding regions, or because of variant coverage issues.

### 4.5. Ethical Statement

The study was conducted in accordance with the Declaration of Helsinki and the National Standard of the Russian Federation “Good Clinical Practice (GCP)” (GOST R52379-2005) and was approved by the Independent Ethics Committee of the NMRC for Therapy and Preventive Medicine (protocol number 07-05/20, dated 26 November 2020). To comply with these regulations, as well as Article 93 of the Federal Law “On the Fundamentals of Health Protection of Citizens of the Russian Federation” (dated 21 November 2011, No. 323-FZ), each participant provided written informed consent for the processing of their personal data. Data from the ESSE-RF study, including the ESSE-FH-RF study and the RPS cohort, were used in the current study. Written informed consent was obtained from each patient as part of their participation in these scientific projects. Data were accessed from 27 November 2020. The database containing clinical, biochemical, and genetic data was deidentified and encrypted to ensure participant confidentiality.

### 4.6. Statistical Analysis

Statistical analyses were conducted using R version 4.4.3 (R Foundation for Statistical Computing, Vienna, Austria) [38]. The distribution of continuous variables was assessed using the Lilliefors test via the nortest package version 1.0-4 [39]. As most continuous variables, including WHtR, CMI, TC, LDL-C, HDL-C, TG, non-HDL-C, remnant cholesterol, and percentile values of HTG PRS and hypercholesterolemia PRS, demonstrated deviations from normality in at least one of the study groups, they were summarized as Me (25th; 75th percentiles). Categorical variables were presented as absolute numbers and percentages. For comparisons of continuous variables between two independent groups, the Mann–Whitney U test was used. In these cases, the pseudomedian difference was estimated using the Hodges–Lehmann method along with its corresponding 95% CI. Comparisons of categorical variables were performed using the two-sided Fisher’s exact test, with the OR and 95% CI estimated. When zero events were observed in one of the comparison groups, Haldane–Anscombe correction was applied, and the OR and its 95% CI were calculated using Wald approximation.

Firth logistic regression was performed to evaluate the association between genetic factors (polygenic risk of HTG, presence of additional variants in lipid-related genes), clinical factors (age, sex, BMI, obesity, WC, WHtR, presence of metabolic syndrome components, glucose metabolism disorders, hypothyroidism), and the presence of FD phenotype (logistf package version 1.26.1 was used [40]). Variables were selected based on their physiological relevance to FD development. Initially, univariable models were used for all selected predictors. Given the clinical and statistical interdependence among several metabolic parameters (e.g., BMI, obesity, WC, and WHtR), including the presence of composite variables (e.g., BMI ≥ 30.0 kg/m^2^ and glucose metabolism disorders as a component of metabolic syndrome), adjustments for multiple comparisons were not applied in the univariable regression analyses in order to avoid overadjustment and the potential loss of clinically important associations.

Then, we created multivariable models. Prior to inclusion in the multivariable analysis, predictors were evaluated for multicollinearity using the variance inflation factor and correlation analysis. Variables with a variance inflation factor of less than 5 and inter-predictor correlations not exceeding 0.7 were included in the final analysis. The final multivariable models included statistically significant genetic and metabolic predictors adjusted for sex, age, and other variables included in the model. Results were presented as OR with 95% CI and *p*-values for each predictor.

Additionally, a linear regression analysis was conducted to assess the associations between FD phenotype-related factors and TG levels across all participants, regardless of group. Two models were presented: univariable unadjusted models for all selected predictors as a first step, and a multivariable model that best explained the variance in TG levels. To assess the model’s predictive performance and avoid overfitting, 5-fold cross-validation was performed using the caret package version 7.0-1 [41]. All key model statistics were reported. A *p*-value of less than 0.05 was considered statistically significant.

Data visualization was carried out using the ggplot2 version 3.5.1 [42], plotly version 4.11.0 [43], VennDiagram version 1.7.3 [44], grid version 4.4.3 (included in base R, [38]), and ComplexUpset version 1.3.3 [45].

#### Statistical Power Analysis

A statistical power analysis was conducted for the main continuous variables of the study using pwr package version 1.3-0 [46] (Table 4). The sample sizes and standard deviations of the analyzed parameters, estimated from the ESSE-Ivanovo study (*n* = 1652) were considered for this analysis. The Type I error rate (α) was set at a two-sided significance level of 0.05, and the Type II error rate (β) was set at 0.2, corresponding to a power of 80%. Based on the sample sizes of 52 patients in the FD group and 19 subjects in the FD variants group, the minimum detectable effect sizes for continuous variables were calculated.

We also calculated Cohen’s d values by dividing the minimum detectable effect size by the corresponding standard deviation obtained from the ESSE-Ivanovo study. All values fell within the range of *d* = 0.76–0.77, which corresponds to medium-to-large or large effect sizes [47]. This suggests that our study was sufficiently powered to detect substantial differences between the FD and FD variant groups. However, smaller yet potentially clinically meaningful effects may remain undetected. Accordingly, nonsignificant results were interpreted with appropriate caution.

### 4.7. Limitations

The study has several limitations. First, we were unable to perform lipoprotein ultracentrifugation or electrophoresis to confirm the FD phenotype. In addition, we could not apply previously developed FD diagnostic algorithms, including apoB levels, to the Russian patients, as described in our prior study [14]. Instead, we defined FD based on the presence of a well-established genetic basis for FD (either the ε2/ε2 *APOE* genotype or rare *APOE* variants causative of FD, the pathogenicity of which were previously demonstrated by our group [1]), combined with TG levels ≥ 1.5 mmol/L. We acknowledge that this approach may reduce the precision of FD classification. However, this methodology and its applicability to a Russian population-based sample have been previously described [14]. Furthermore, another study has shown that a TG cutoff of ≥1.5 mmol/L may help to identify subjects with less pronounced FD phenotypes [48]. Second, we did not recalculate HDL-C levels for patients on regular LLT because of a lack of reliable data on the impact of different LLTs on these parameters. Third, larger cohort or population-based studies specifically examining the genotype-phenotype relationship in patients with the AD form of FD are recommended. These studies would allow for a more detailed understanding of the FD phenotype and its modifiers in this subgroup.

## 5. Conclusions

Age, polygenic burden for HTG, and the presence of metabolic syndrome components were independently associated with the FD phenotype. Additionally, polygenic HTG, rare lipid-related variants beyond *APOE*, and glucose metabolism disorders were significantly linked with TG levels, explaining 30% of TG variance in a cross-validated model. These findings underscore the complex interplay of genetic and metabolic factors in FD and support the integration of TG-PRS and metabolic control into personalized risk assessment and management strategies for FD patients.

## Figures and Tables

**Figure 1 ijms-26-07376-f001:**
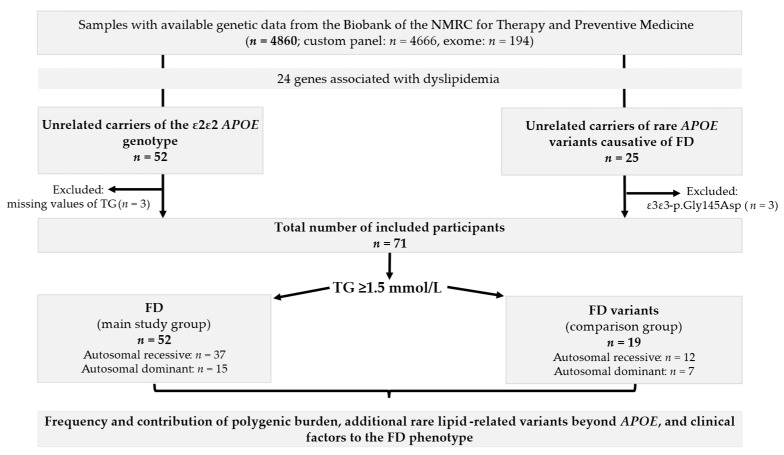
Study design. FD—familial dysbetalipoproteinemia.

**Figure 2 ijms-26-07376-f002:**
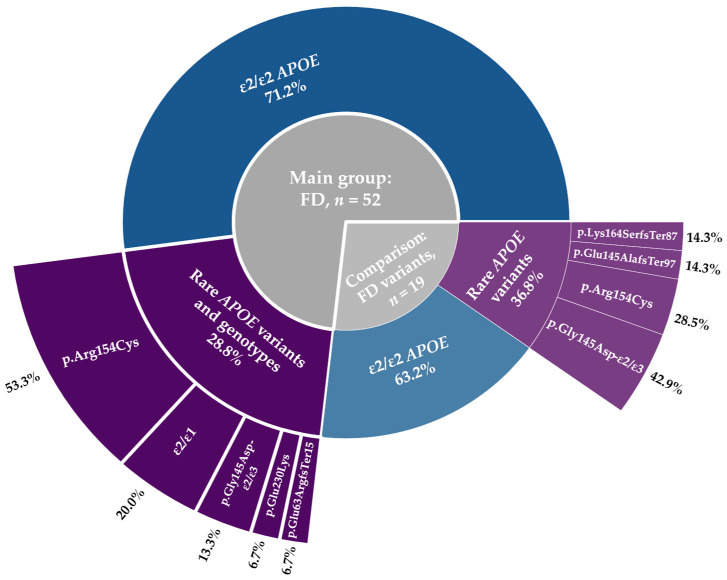
Genetic spectrum of the study groups. The inner gray circle separates the main FD group (*n* = 52) from the comparison group with FD variants (*n* = 19). The outer ring shows the proportion of ε2/ε2 cases (blue) and various rare *APOE* variants (purple) within each group. Percentages within each group are indicated.

**Figure 3 ijms-26-07376-f003:**
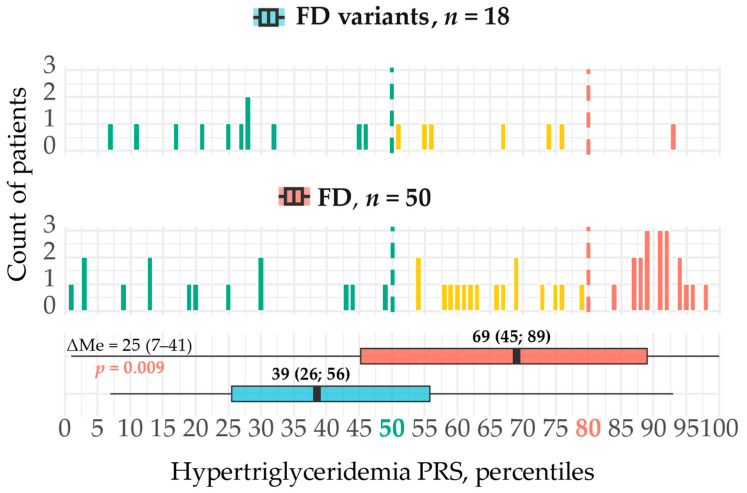
Distribution of subjects with FD variants and patients with FD by HTG PRS percentiles. Bar heights represent the number of subjects per percentile. Bars are color-coded by percentile category: green (<50th percentile), yellow (50th–80th), and red (>80th). Dashed vertical lines mark the 50th (green) and 80th (red) percentiles. Horizontal boxplots below panels summarize HTG PRS percentile distributions: central lines represent the median, box limits represent upper and lower quartiles, and horizontal lines represent 1.5 times the quartile range. The median (25th; 75th percentiles) is labeled above each boxplot. Group differences were analyzed using the Mann–Whitney U test; ΔMe and 95% CI were estimated via the Hodges–Lehmann method. FD—familial dysbetalipoproteinemia; ΔMe—median difference; PRS—polygenic risk score.

**Figure 4 ijms-26-07376-f004:**
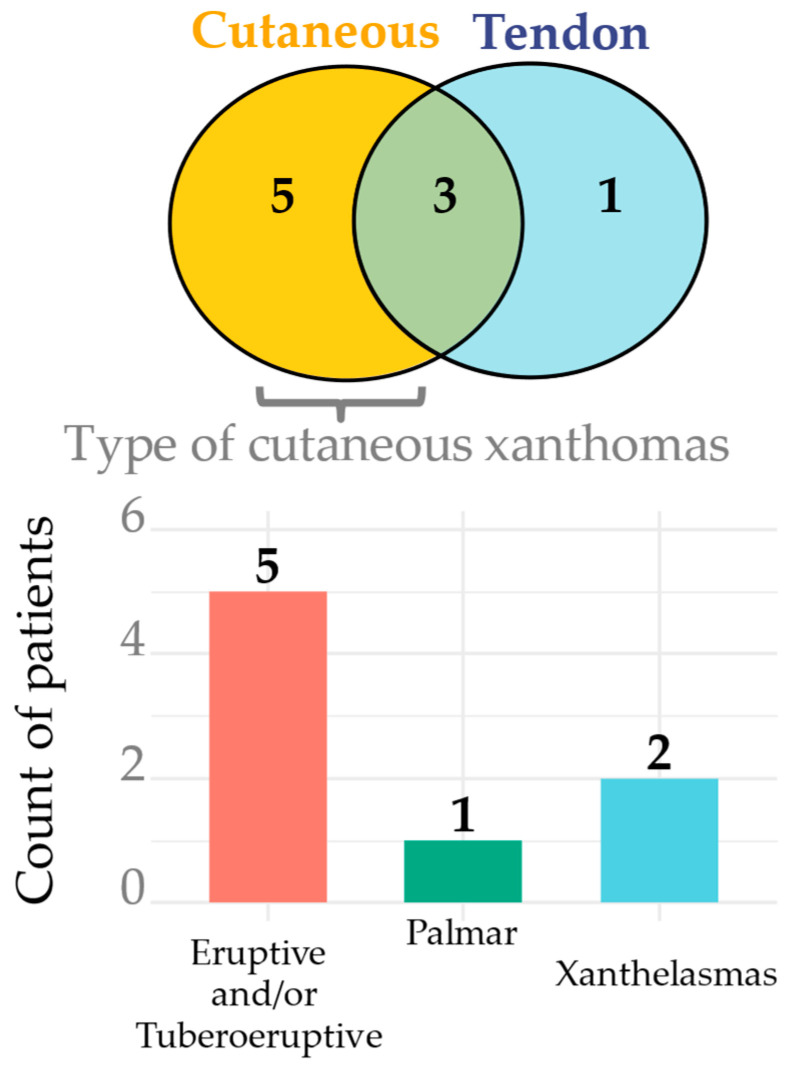
Xanthomas in FD patients.

**Figure 5 ijms-26-07376-f005:**
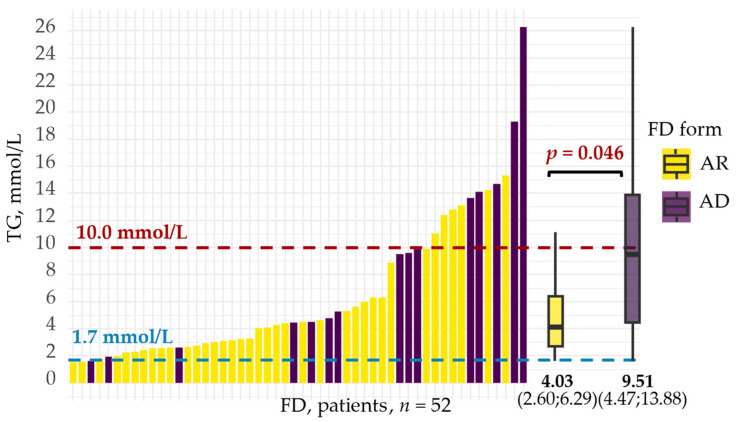
TG level distribution among patients with AR and AD forms of FD. Bar heights indicate TG levels. Yellow indicates patients with AR FD, while purple indicates patients with AD FD. Boxplots summarize TG levels: central lines represent the median, box limits represent upper and lower quartiles, and vertical lines represent 1.5 times the quartile range. The median (25th; 75th percentiles) is labeled under each boxplot. Group differences were analyzed using the Mann–Whitney U test. AD—autosomal dominant; AR—autosomal recessive; FD—familial dysbetalipoproteinemia; TG—triglycerides.

**Figure 6 ijms-26-07376-f006:**
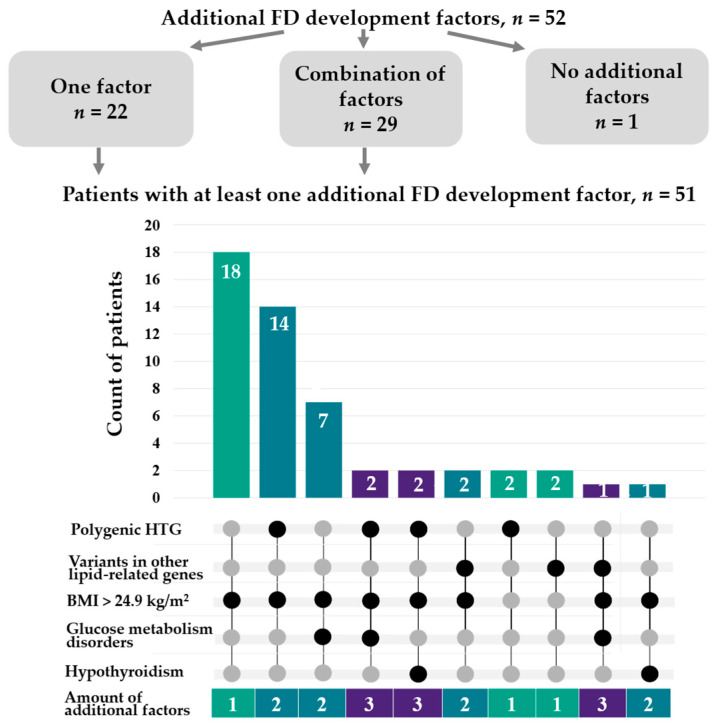
Combinations of polygenic HTG, pathogenic variants in other lipid-related genes, increased body weight, glucose metabolism disorders, and hypothyroidism in FD patients. The chart illustrates the combinations of these factors among FD patients. Black dots indicate the factors included in a combination. The bar graph shows the number of patients with each combination. Colors indicate patients with the same number of additional factors. BMI—body mass index; FD—familial dysbetalipoproteinemia; HTG—hypertriglyceridemia.

**Figure 7 ijms-26-07376-f007:**
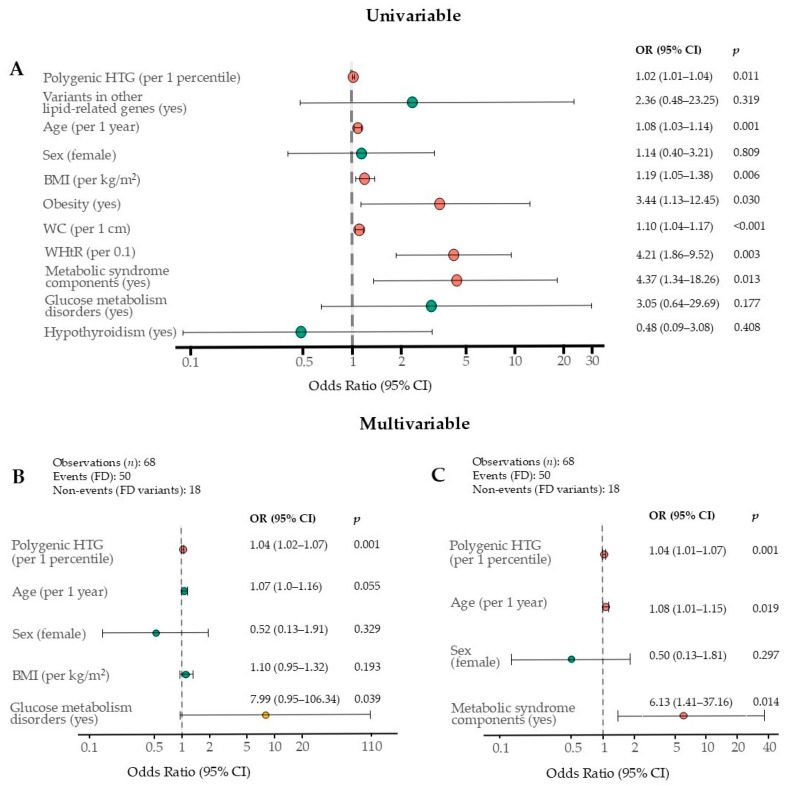
Independent FD phenotype factors identified by univariable (**A**) and multivariable (**B**,**C**) Firth logistic regression: (**A**) This forest plot shows the results of the univariable unadjusted analyses. The number of observations for all variables, except polygenic HTG, WC, WHtR, and hypothyroidism, was 71 (events (FD) = 52, non-events (FD variants) = 19); for polygenic HTG, 68 (50 and 18, respectively); for WC and WhTR, 58 (42 and 16, respectively); and for hypothyroidism, 69 (51 and 18, respectively); (**B**) multivariable model I; (**C**) multivariable model II. Red dots indicate significant associations; green dots indicate nonsignificant associations; yellow dots indicate significant *p*-values with wide CIs including 1.00. BMI—body mass index; CI—confidence interval; FD—familial dysbetalipoproteinemia; HTG—hypertriglyceridemia; OR—odds ratio; WC—waist circumference; WHtR—waist-to-height ratio.

**Table 1 ijms-26-07376-t001:** Pathogenic variants in additional lipid-related genes and their combinations with *APOE* variants.

Study Group	Carriers No.	Additional Variant 1, Gene/HGVSp/Zygosity	Additional Variant 2, Gene/HGVSp/Zygosity	*APOE* Genotype	Rare *APOE* Variant HGVSp/Zygosity
FD	1	*LDLR/*p.Ser554Leu/het	-	ε2/ε2	-
	2	*LDLR/*p.Ser586Pro/het	-	ε2/ε2	-
	3	*LPL/*p.Asp202Asn/het	-	ε2/ε2	-
	4	*LPL/*p.Asp202Asn/het	*LPL/*p.Tyr233Cys/het	ε3/ε3	p.Arg154Cys/het
	5	*LMF1/*p.Tyr439Cys/homo	-	ε2/ε2	-
Comparison	1	*LDLR/*p.Gly592Glu/het	-	ε2/ε3	p.Gly145Asp/het

HGVSp—Human Genome Variation Society protein sequence name; FD—familial dysbetalipoproteinemia.

**Table 2 ijms-26-07376-t002:** Comparison of clinical and biochemical data between FD patients and subjects with FD variants.

Parameter	All Subjects (*n* = 71)	Carriers of the ε2/ε2 Genotype and Rare *APOE* Variants (*n* = 19)	Patients with FD (*n* = 52)	*p*-Value ^1^	OR (95% CI) ^2^/ ΔMe (95% CI)
Men, *n* (%)	32 (45.1)	9 (47.4)	23 (44.2)	1.0	1.1 (0.3–3.7)
Age, years, Me (Q1; Q3)	50 (40; 56)	40 (31; 52)	51 (46; 58)	0.004	11 (4–18)
Smoking (current or former smokers), *n* (%)	32 (45.1)	8 (42.1)	24 (46.2)	0.794	1.2 (0.4–4.0)
Hypertension, *n* (%)	42 (59.2)	8 (42.1)	34 (65.4)	0.104	2.6 (0.8–8.8)
Metabolic Factors					
BMI, kg/m^2^, Me (Q1; Q3)	28.7 (25.4; 31.7)	24.9 (22.8; 28.8)	29.5 (26.9; 31.9)	0.001	4.1 (1.7–6.4)
WC, cm, Me (Q1; Q3)	94.0 (84.2; 100.2) *n* = 58	81.0 (75.4; 95.5) *n* = 16	95.0 (90.0; 104.0) *n* = 42	0.002	13.0 (6.0–21.0)
WHtR, Me (Q1; Q3)	0.6 (0.5; 0.6) *n* = 58	0.5 (0.4; 0.6) *n* = 16	0.6 (0.6; 0.6) *n* = 42	0.001	0.1 (0.04–12.7)
CMI, Me (Q1; Q3)	1.300 (0.622; 3.410) *n* = 58	0.337 (0.176; 0.523) *n* = 16	2.160 (1.050; 5.150) *n* = 42	<0.001	1.823 (1.030–3.670)
Glucose metabolism disorders, *n* (%) ^4^	11 (15.5)	1 (5.3)	10 (19.2)	0.267	4.2 (0.5–196.0)
Metabolic syndrome components, *n* (%) ^5^	40 (56.3)	3 (15.8)	37 (71.2)	<0.001	13 (3–77)
Hypothyroidism, *n* (%)	5 (7.2) *n* = 69	2 (11.1) *n* = 18	3 (5.9) *n* = 51	0.600	0.5 (0.1–6.6)
CHD					
CHD, *n* (%)	14 (19.7)	0	14 (26.9)	0.015	14.7 (0.8–259.4) ^3^
Age at onset of CHD, years, Me (Q1; Q3)	42 (37; 49)	—	42 (37; 49)	—	—
Pretreatment Lipid Levels					
TC, mmol/L, Me (Q1; Q3)	5.63 (4.23; 9.67) *n* = 67	4.15 (2.87; 4.65)	6.92 (4.99; 10.83) *n* = 48	<0.001	2.93 (1.67–5.25)
LDL-C, mmol/L, Me (Q1; Q3)	2.39 (1,66; 3.31) *n* = 61	1.66 (0.99; 2.32)	2.95 (2.11; 3.55) *n* = 42	0.001	1.13 (0.55–1.74)
HDL-C, mmol/L, Me (Q1; Q3)	1.09 (0.89; 1.44)	1.36 (1.17; 1.69)	1.03 (0.84; 1.31)	0.002	−0.33 (−0.57; −0.14)
TG, mmol/L, Me (Q1; Q3)	3.01 (1,47; 6.15)	0.9 (0.70; 1.29)	4.47 (2.65; 9.67)	<0.001	3.52 (2.24–5.05)
Non-HDL-C, mmol/L, Me (Q1; Q3)	4.28 (2.69; 8.33) *n* = 57	2.43 (1.65; 3.27)	5.69 (4.01; 9.98) *n* = 38	<0.001	3.26 (1.92–6.08)
Remnant cholesterol, mmol/L, Me (Q1; Q3)	1.34 (0.71; 2.02) *n* = 50	0.65 (0.46; 0.91)	1.87 (1.25; 2.59) *n* = 31	<0.001	1.17 (0.70–1.70)

^1^ *p*-values indicate differences between two groups. The Mann–Whitney U test was used for continuous variables; ΔMe and 95% CI were estimated using the Hodges–Lehmann method. For categorical variables, *p*-values and OR with 95% CI were obtained using the two-sided Fisher’s exact test. ^2^ Results are presented for the FD group compared with the FD variants group. ^3^ Because of the absence of CHD cases in the FD variants group, Haldane–Anscombe correction was applied. OR and 95% CI were calculated using Wald approximation. ^4^ Glucose metabolism disorders included diabetes mellitus, impaired glucose tolerance, and impaired fasting glucose. ^5^ Metabolic syndrome components included at least two of the following: hypertension, BMI ≥ 30.0 kg/m^2^, or glucose metabolism disorders. BMI—body mass index; CI—confidence interval; CHD—coronary heart disease; CMI—cardiometabolic index; FD—familial dysbetalipoproteinemia; HDL-C—high-density lipoprotein cholesterol; LDL-C—low-density lipoprotein cholesterol; ΔMe—median difference; Non-HDL-C—non-high-density lipoprotein cholesterol; OR—odds ratio; TC—total cholesterol; TG—triglycerides; WC—waist circumference; WHtR—waist-to-height ratio.

**Table 3 ijms-26-07376-t003:** The multivariable model that best explained the variance in TG levels.

Predictor	β Coefficient (SE)	95% CI	*p*-Value
Polygenic HTG, per 1 percentile	0.05 (0.02)	0.01–0.09	0.011
Variants in other lipid-related genes (yes)	7.0 (1.70)	3.60–10.41	< 0.001
Glucose metabolism disorders (yes)	3.62 (1.52)	0.58–6.66	0.020
Initial model statistics: R^2^ = 0.30, Adjusted R^2^ = 0.27, Residual SE = 4.50 (df = 64), *p* < 0.001, observations number = 68			
Fivefold cross-validation model statistics ^1^: R^2^ = 0.30, Residual SE = 4.74, MAE = 3.48, observations number = 68			

^1^ Cross-validated metrics were derived using fivefold cross-validation with the caret package in R. CI—confidence interval; HTG—hypertriglyceridemia; MAE—mean absolute error; SE—standard errors.

**Table 4 ijms-26-07376-t004:** Statistical power analysis.

Parameter	Standard Deviation in ESSE-Ivanovo (*n* = 1652)	Minimum Detectable Effect Sizes	Cohen’s *d*
HTG polygenic risk, score	0.21	0.16 ^1^	0.76
Hypercholesterolemia polygenic risk, score	0.34	0.26 ^2^	0.76
BMI, kg/m^2^	5.90	4.49	0.76
TG, mmol/L	1.04	0.79	0.76
LDL-C, mmol/L	1.15	0.88	0.77
HDL-C, mmol/L	0.34	0.26	0.76

^1^ FD (*n* = 50) and FD variants (*n* = 18). ^2^ FD (*n* = 50) and FD variants (*n* = 19). Statistical power 80.0%. Cohen’s *d* was calculated as minimum detectable effect size/standard deviation. BMI—body mass index; HDL-C—high-density lipoprotein cholesterol; HTG—hypertriglyceridemia; LDL-C—low-density lipoprotein cholesterol; TG—triglycerides.

## Data Availability

The data used in this study, including individual genotype information, cannot be publicly disclosed according to the rules of the Ethics Committee of the National Medical Research Center for Therapy and Preventive Medicine. Deidentified data will be provided upon reasonable request by the corresponding author, Anastasia Blokhina (blokhina0310@gmail.com), or by the Ethics Committee of the National Medical Research Center for Therapy and Preventive Medicine (phone number +74995536810, secretarynec@gnicpm.ru). Proposals will be reviewed and approved by the investigators, local regulatory authorities, and the Ethics Committee of the National Medical Research Center for Therapy and Preventive Medicine. Once the proposal is approved, data can be transferred through a secure online platform after signing a data access agreement and a confidentiality agreement.

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
