# Peer review of "Genetic and Metabolic Factors of Familial Dysbetalipoproteinemia Phenotype: Insights from a Cross-Sectional Study"

_ijms, 2025, doi:10.3390/ijms26157376_

Round 1
Reviewer 1 Report
Comments and Suggestions for Authors
The study entitled “Genetic and Metabolic Factors of Familial Dysbetalipoproteinemia Phenotype: Insights from the Cross-Sectional Study” very clearly elucidates the complex interplay of genetic and metabolic factors influencing FD and demonstrating how elevated polygenic risk and metabolic disorders contribute to its phenotype and emphasizing the potential for personalized patient management. The following ponts shall be considered before considering the manuscript for publication.
The abstract is not with sufficient details and would benefit significantly from revision to improve clarity and more precise details about the sampling, Polygenic Risk Score, etc. The lack of FD phenotype should be mentioned in the abstract. The conclusion should also be more concise and clearly highlight the main contributions of the study.
Introduction of the study is not reflecting the appropriate background of the study. The introduction could more add the unique contribution in investigating both AR and AD forms. Briefly add tge comprehensive genetic and clinical factor analysis in FD subtypes. Specific factors like polygenic risk and metabolic syndrome components should be be included to link the phenotype as it is missing and major limiting factor.
Authors should elaborate how the study (using a large genetic dataset) specifically addresses in identifying independent associations of the phenotype.
Integration of genetic risk scores for personalized FD management shall be discussed.
Figure 2 of Genetic spectrum of the study groups is not self explanatory. Should revise and ensure the colors and its association included in the legend and figure.
For better alignment with the study consider emphasizing the quantitative contribution to TG variance in conclusion. More clearly link the integration of risk scores and metabolic control to personalized FD management in the conclusion as powerfully stated in your abstract.
Authors acknowledged several limitations particularly the support on a TG threshold for FD definition without lipoprotein ultracentrifugation or apoB levels. While it is added a brief discussion on how these methodological constraints might impact the generalizability or precision of phenotype association for the cohortt especially regarding the identified genetic and metabolic factors will be necessary.
Author should elaborate the presentation of the calculated minimum detectable effect sizes in the Statistical Power Analysis.
Author should briefly state reasons for the exclusion of the 2-3 patients for whom PRS was not calculated.
Comments on the Quality of English LanguageEnglish could be improved to more clearly
Reviewer 2 Report
Comments and Suggestions for Authors
The authors conducted a cross-sectional study on genetic and metabolic factors associated with Familial Dysbetalipoproteinemia (FD) development in the Russian population. The study's findings highlight the complex interplay between genetic and metabolic influences on FD and support the use of polygenic risk scores for hypertriglyceridemia as a predictor of FD risk. The study seems to be well-conducted and the manuscript is well-written. However, I have some suggestions for improving the organization of the content, which would enhance the readability and quality of the presentation.
- Lines 70-73: Provide some references and complete the paragraph.
- Line 88. p.Arg154Cys (n=10) - were there any relatives in the sample?
- Line 90 and beyond. For the homozygous variant, only one allele (present) needs to be mentioned.
- Statistical analysis lacks correction for multiple comparisons. Although most of the data clearly retain their significance after adjusting for multiple comparisons, it is still important to include an FDR (false discovery rate) correction in the statistical analysis.
- Polygenic Risk Score. Please provide brief information about the PRS panel in the supplementary materials.
- Line 253: Did the authors perform cross-validation on their model? Could the authors please provide the results of the final model after the cross-validation process?
- The authors conducted the analysis on a Russian population. Have they performed PCA analysis to exclude patients of non-Russian ethnicity? Please provide these data and exclude patients from other ethnic groups. These results may be included in the supplementary materials.
- Line 445: Continuous variables were summarized using the median (25th percentile; 75th percentile). Could the authors please explain the reasoning behind choosing this method of data presentation? Have the authors analyzed the data distribution?
- In the conclusion, the authors should include a brief paragraph on how the results of their study contribute to achieving the goal of “early identification of atherogenic factors and reducing the burden of cardiovascular disease” (lines 71-73). They should also discuss how their findings can be applied in clinical settings to improve patient care.
- The results were obtained in a Russian population, which, in terms of global superpopulations, is a population with European ancestry. Were there any new findings in the study beyond the population factor? Please include this information in the conclusion.
- Based on the study results, what are some possible scientific perspectives?
Round 2
Reviewer 1 Report
Comments and Suggestions for Authors
Revised MS can be accepted
Comments on the Quality of English LanguageNeed English editing
Reviewer 2 Report
Comments and Suggestions for Authors
All my comments have been taken into account.